# Discovering smart cities' potential in Kazakhstan: A cluster analysis

**Marat Urdabayev[1]☯, Anel Kireyeva[2]☯, Laszlo Vasa [3]☯\*, Ivan Digel[2]☯, Kuralay Nurgaliyeva[4]☯, Akan Nurbatsin[4]☯**

1 Department of Economics, Al-Farabi Kazakh National University, Almaty, Kazakhstan, 2 Department of information and implementation of research results, Institute of Economics of the Ministry of Science and Higher Education of RK, Almaty, Kazakhstan, 3 Széchenyi István University, Győr, Hungary, 4 Department of Science, Kenzhegali Sagadiyev University of International Business, Almaty, Kazakhstan

☯ These authors contributed equally to this work.
\* laszlo.vasa@ifat.hu

**Data Availability Statement:** All relevant data are within the manuscript and its Supporting Information files.

**Funding:** This research has funded by the Science Committee of the Ministry of Science and Higher

## Abstract

The potential for developing smart cities in Kazakhstan is evaluated using cluster analysis. Built on previous research focused on clustering the regions of Kazakhstan, this study applies the same method to the cities of the country. The analysis uses indicators related to human capital, infrastructure, education, information technology, production, and other factors to assess the potential of each city. The clustering is performed using Single Linkage, Complete Linkage, and Ward's methods. The results show that Almaty and Astana are the cities with the highest potential for becoming smart cities. Aktobe is identified as a city with distinctive features that may help or hinder its development as a smart city. The remaining cities are clustered into two groups, with one group having the potential to catch up and maintain the trend of developing smart cities, while the other group is less suitable for starting smart city projects and may require more investment per capita. The study highlights the deep regional inequality affecting the potential to successfully develop and manage smart cities in Kazakhstan. The analysis also reveals some limitations and challenges in the data and variables used, including the lack of data for some variables and the difficulties in "translating" some factors and indicators into quantitative variables for clustering. The study concludes that future research should address these challenges and consider clustering inside certain regions to focus on their unique features. The study recommends launching pilot projects in small cities, with the most successful practices then scaled and implemented in the core smart cities and possibly Aktobe, if it manages to use its advantages to compensate for risks. Overall, this study provides insights into the potential of smart city development in Kazakhstan and can inform policymakers in their efforts to support smart city projects in the country.

## Introduction

Urbanization and the rapid development of technology have led to growing interest in the concept of smart cities, which aim to integrate information and communication technologies

Education of the Republic of Kazakhstan (Grant " Development Strategy of Kazakhstan Regional Potential: Assessment of Socio-Cultural and Economic Potentials, Roadmap, Models and Scenarios Planning " No. BR18574240)." The funders had no role in study design, data collection and analysis, decision to publish, or preparation of the manuscript.

**Competing interests:** The authors have declared that no competing interests exist.

(ICT) to enhance the quality of life, economic development, and sustainability of urban areas. Smart cities represent a promising approach to address the challenges of urbanization, including congestion, pollution, and resource scarcity. With an increasing number of countries exploring the potential benefits of smart city development, there is a pressing need to understand how different cities can adapt and implement these concepts.

In this context, the present study focuses on Kazakhstan, a country with a diverse range of cities and unique challenges. Despite the growing interest in smart cities globally, little attention has been paid to Kazakhstan, leaving a gap in understanding the potential for smart city development in the country. This paper aims to contribute new insights about Kazakhstani cities by using cluster analysis. Using available data on various socioeconomic, environmental, technological, and political factors, the study seeks to group Kazakh cities into clusters with similar potential for smart city development. The results of this research will provide insights into the current state of Kazakh cities and offer recommendations for policymakers and urban planners to prioritize and strategize the development of smart cities in Kazakhstan.

Smart city development involves a holistic approach, taking into account various interconnected aspects such as transportation, energy, waste management, and public services, among others. The integration of ICT solutions is critical to achieving these goals, as it enables real-time monitoring and data-driven decision-making. Consequently, the transformation towards smart cities often requires significant investments in infrastructure, technology, and human capital.

Kazakhstan, with its unique geographical, political, and economic characteristics, presents a compelling case study for smart city development. The country is vast, with diverse urban areas that range from densely populated metropolitan centers to smaller cities and towns with varying levels of development. As the largest landlocked country in the world, Kazakhstan faces distinct challenges related to transportation and logistics, making the implementation of smart city concepts even more critical for enhancing connectivity and economic development. Furthermore, the country's rich natural resources, including oil and gas reserves, have fueled economic growth, but have also contributed to environmental challenges that need to be addressed through sustainable urban planning.

The government of Kazakhstan has acknowledged the importance of smart city development and has taken steps to support the digital transformation of its urban areas. The "Digital Kazakhstan" program, launched in 2017, outlines a national strategy for digitalization, focusing on enhancing the quality of public services, strengthening the digital infrastructure, and fostering innovation. While these initiatives provide a strong foundation for smart city development, it is crucial to understand the specific needs and potential of individual cities to tailor strategies accordingly.

In addition to the national context, it is essential to consider regional and local factors when assessing the potential for smart city development in Kazakhstan. The country's regions exhibit varying levels of development, access to resources, and environmental conditions, which can significantly influence the feasibility and effectiveness of smart city initiatives. By incorporating regional and local data into the cluster analysis, this study aims to capture these differences and provide a more accurate assessment of the potential for smart city development across Kazakhstan.

This paper builds on previous research, the work of Digel et al. [1], which used cluster analysis to assess the potential of smart cities in different regions of Kazakhstan. However, the main shortcoming of the previous study was that it did not directly analyze cities—only the administrative regions of the country. Since we are talking about smart cities and not smart regions, it makes sense to consider each city separately. The current study goes deeper by focusing on cities of Kazakhstan and employing a more detailed set of variables. In addition,

the previous study had a poorly justified set of variables, and the present study also addressed this deficiency.

The cluster analysis methodology employed in this study allows for a flexible and robust approach to grouping cities based on their similarities and differences. By exploring various clustering techniques and determining the optimal number of clusters, the research aims to provide meaningful and actionable insights for policymakers and urban planners. The results of the cluster analysis will not only highlight the cities with the highest potential for smart city development but also identify those that may require additional support and investments to overcome specific challenges.

The factors used to form a set of variables are derived from Frazer's comprehensive framework [2] for smart city development, which emphasizes the importance of addressing all aspects of urban life to achieve sustainable growth. By collecting and analyzing data on these factors for cities across Kazakhstan, the study identifies patterns and group cities into clusters with similar potential for smart city development.

It is essential to note that the present study is not without limitations. Data availability is a significant constraint, with some cities lacking complete data for all selected variables. This limitation has led to the exclusion of certain cities from the analysis and may affect the overall results. Additionally, the process of translating Frazer's framework [2] into quantitative variables for clustering presents challenges, as some factors may not be fully represented or captured by the available data. Despite these limitations, this study produces results worth a discussion and widens the research about smart cities in Kazakhstan. The results will not only serve as a resource for policymakers and urban planners but will also contribute to the growing body of literature on smart city development in emerging economies.

## Literature review

There are various definitions of a smart city, where initially they were based on the use of technologies and innovation in the management process of a city. Despite the discussion of various concepts and theories, there is no consensus on a clear definition of the term smart city. Hollands [3]investigated smart cities, identifying important issues: assumptions about a smart city as a festive label, that this label is more of a marketing hype than a practical factor of infrastructure change, and the term itself, carrying an uncritical connotation of development. Leydesdorff and Deakin [4] emphasized that smart cities are a process of cultural reconstruction based on policy, academic leadership, and corporate strategy of their management.

Moreover, scientists underline that smart initiatives are developed in accordance with the characteristics specific to the region [5]. Therefore, definitions are developed according to specific characteristics of a city, which are also regarded as components. Nam and Pardo [5] identified technologies, population and communities. Alawadhi et al. [6] divided the components into two groups. The first, included government at the state and regional levels, population, level of infrastructure and economy development and natural environment. The second also included technologies, organizations and state policy. Dameri [7] outlined four components: human capital, area, government and technologies. Bibri and Krogstie [8] divided the components into two main groups: management of technologies and innovation incorporation and human capital.

Recent studies show that current concept of a smart city is based on the principles of building sustainable development capacity of urban areas provided through incorporation of digital solutions in the management process of a city [9]. They affect and thus increase the happiness level of the population by improving quality of life through several indicators: environment pollution reduction, employment and education conditions which consider needs of the society and infrastructure development [10].

In general, smart cities widely use information and communication technologies to help large cities create their competitive advantages. Lopes [11] outlined two major factors as use of technologies and smart governance. Additionally, the study included context analysis that was based on five groups of factors including political, social, cultural, organizational and technological and allowed analysing possibilities for incorporation of smart governance. It is implemented through smart solutions, which include main factors for smart cities development, as electronic governance, services provision, consultations and ICT availability [11].

Radziejowska and Sobotka [12] emphasized that human capital is the main factor for smart cities development. Moreover, they state that the level of urban population competence and interest in the incorporation of smart technologies affect the implementation of smart solutions. The importance of society readiness to smart cities is explained by the security factor as individuals and society overall aim at reducing the risk of personal data accessibility by third parties [13, 14]. Singh and Singla [15] identified population quality of life, smart solutions, in particular e-services and economy development as crucial factors for smart cities development.

Therefore, the effect of smart cities is regarded as controversial as it can have positive or negative results. Existing studies direct their attention to the characteristics of a city and thus differ in selected factors that affect smart cities development.

A similar methodological approach to evaluating smart cities using cluster analysis was tested in the work of Carmen Cantuarias-Villessuzanne, Weigel Romena, and Blain Jeffrey [16], where they analyzed the "smart" strategies of European cities and developed a clustering of smart cities based on the activities carried out by the cities [16]. Lytras, Visvizi, and Sarirete investigated the clustering of smart city services [17]. Xiang et al. [18] studied data stream processing applications in a smart city using a cluster analysis approach. Indonesian scientists have also conducted similar research. Muntean conducted a study using cluster analysis to predict and solve the parking space occupancy problem in the smart city of Birmingham. Her approach involves first grouping the dataset to get the relevant periods throughout the day, and then predicting the data in these clusters [19]. Safitri et al. [20] conducted a cluster analysis of smart city regions in Banda Aceh, Indonesia. This study identified the similarities in the characteristics of each object in the Aceh regions. In addition, Srinivas and Hosahalli studied a MapReduce distributed computing environment based on clustering using K-means evolutionary computation for an IoT-based smart city [21].The use of cluster analysis in this work evokes associations with innovation clusters, since the concept of smart city largely relies on innovation implementation. Brakman and van Marrewijk believe that the effect of clusters in cities lies in their impact on the national and regional economies of cities, as their effect has a favorable impact on the existing mechanism in agglomerations [22] Van Klink and de Langen believe that the city cluster must go through a mandatory life cycle of innovative product development. The city cluster involves the emergence of small companies that develop innovative products in smart cities [23]. Also, in their opinion, the cluster grows with the arrival of new companies and highly qualified specialists who implement innovative "smart" projects. The specialization of the cluster increases until it reaches maturity at a certain point. If companies in the cluster fail to adapt to changing economic conditions, a decline occurs.

According to Köcker and Müller, the main goals of cluster policy are to increase labor productivity, speed up the development of innovative products, and improve the competitiveness of small and medium-sized enterprises in the region [24]. Noiva, Fernandez, and Weskott analyzed a dataset from 142 cities, which included the annual water consumption per capita. Using these indicators of urban water supply and consumption, they performed a hierarchical cluster analysis to identify relative similarities and distances between the 142 cases [25]. Kubina, Šulyová and Vodak compared standards, implementation, and cluster models for

smart cities in North America and Europe, using cluster analysis [26]. Héraud and Müller studied the interaction between smart cities and innovation clusters, as well as people involved in technology clusters, research centers, factory laboratories, living labs, etc. [27].

Nazarova and Demianenko conducted a cluster analysis of regions in Ukraine. According to the results of the cluster analysis, the regions were grouped into six clusters. The dynamics of the quantitative distribution of Ukrainian regions by the selected clusters were also analyzed. The study identified cores with a constant composition of regions and presented characteristics of each cluster [28].

The analysis of smart cities in Kazakhstan, as well as the cluster analysis of regions and cities, were also conducted in Kazakhstan's research. Urdabayev and Turgel evaluated the applicability of the "Smart Aqkol" case in the development of smart cities in other cities of Kazakhstan [29].There is the "WeAlmaty" project, which was implemented by the British Council, the government of Almaty city, the "Almaty Urban Development Center" JSC, and the Kazakh-British Technical University.

Aralbaeva and Berikbolova have examined the cluster analysis of the regions of Kazakhstan in terms of their level of innovative development [30]. According to them, one of the effective methods of managing the possibilities of sustainable development of cities in the cluster policy of Kazakhstan is the formation of interconnected forms of suppliers and universities. Cluster policy is an effective form of relationships in the internal environment of the regions of the Republic of Kazakhstan, which has recently become a dominant component. The main task of cluster policy is to create favorable conditions for the development of regional economy, depending on the category of the cluster and the strategies of the regions [30]. Satpayeva, Kireyeva, Kenzhegulova and Yermekbayeva [31] have proposed methodological tools based on a systemic approach using economic and statistical methods and the concept of 5Ms. Additionally, Mussabalinaand Kireyeva [32] believe that the topic of cluster development in Kazakhstan deserves special attention, since attempts have been made in the country to support and develop cluster policy aimed at the socio-economic development of the state of Kazakhstan and its regions.

Based on the literature review, it can be concluded that there are studies dedicated to the problems of formation and management of the development of smart cities. Some works are devoted to the use of cluster analysis to determine the level of development of regions and the grouping of cities. Thus, some studies are related to the analysis of smart cities and existing clusters in Kazakhstan, as well as works in which cluster analysis is used as a research method on similar topics. However, there are very few works aimed at using cluster analysis to study the regional environment with the aim of forming a smart city. Moreover, there have been no works in which cluster analysis was used to determine the potential of Kazakhstan's regions for the development of smart cities.

The cluster policy for the development of clusters of smart cities and their management allows for the increase of innovative activity by strengthening small and medium-sized enterprises, focusing on the common strategic goal and innovation. Therefore, the aim of this article is to study the cites of Kazakhstan and identify the best potential locations for the development of smart cities based on cluster analysis.

## Methods and data

The method of this research is agglomerative clustering. Agglomerative clustering is a hierarchical clustering method used to group similar data points together. This method starts with each data point forming its own cluster and then progressively merges them based on a similarity criterion until only one cluster remains. The agglomerative clustering method operates

by computing a proximity measure between pairs of clusters, which indicates how similar or dissimilar they are. The proximity measure can be based on various distance metrics such as Euclidean distance or correlation coefficient. Based on this proximity measure, the algorithm merges the two closest clusters into a single new cluster. The process of computing the proximity measure and merging the clusters is repeated iteratively until all data points are in a single cluster.

The agglomerative clustering method has several advantages, including its simplicity, flexibility, and ability to handle large datasets. However, it can be computationally expensive for large datasets, and the results may be sensitive to the choice of linkage criterion. Since the dataset of the present paper is not big, the only problem that remains is the sensitivity to the choice of linkage criterion, which is solved by using several criteria for the analysis.

There are different linkage criteria that can be used to measure the proximity between clusters. Those used in this paper are single linkage ("nearest neighbor"), complete linkage ("furthest neighbor"), and Ward's clustering methods.

Ward first described his method in 1963 [33]. It is is a popular linkage criterion used in agglomerative clustering that aims to minimize the within-cluster variance. The basic idea behind Ward's method is to find the pair of clusters whose merger results in the smallest increase in the sum of squared deviations from the mean of the combined cluster.

The function D(X,Y) that calculates the distance between clusters measures the increase in the "sum of squared errors" (SSE) after merging two clusters.

$$D(X, Y) = ESS(XY) - [ESS(X) + ESS(Y)] \qquad (1)$$

where ESS(.) takes the form:

$$ESS(X) = \sum_{i=1}^{N_X} \left| x_i - \frac{1}{N_x} \sum_{j=1}^{N_X} x_j \right|^2 \qquad (2)$$

where NX is the number of elements in the cluster, xi and xj are elements of the cluster. The goal of the method is to choose such a sequence of clustering steps that minimizes D(X,Y) (increase in SSE at each step).

One of the main advantages of Ward's method is that it produces compact, spherical clusters with relatively uniform sizes, which can be useful for certain types of datasets. However, Ward's method can be sensitive to outliers, as it tends to prioritize merging small clusters with other small clusters, even if they are far apart from each other.

Single and complete linkage methods work differently from Ward's method. They still start with as many clusters as there are observations and end up with only one, but the criteria for merging are different from Ward's method. Both methods use Euclidean distance:

$$D_{ij} = \sqrt{\sum_i (a_i - b_i)^2} \qquad (3)$$

where ai and bi are the selected variables of the corresponding observations A and B. At each step, two clusters with the smallest distance are merged. The methods differ in how the new distance is calculated. For single linkage, the distance between the two closest points of the two clusters is calculated (i.e. nearest neighbors), and for complete linkage, the distance between the two farthest points of the two given clusters is calculated (i.e. farthest neighbors) [34].Single linkage tends to produce long, branching clusters that are sensitive to noise and outliers. Complete linkage tends to produce compact, spherical clusters with relatively uniform sizes, but it can be sensitive to clusters with varying densities. In other words, the three applied methods should complement each other. Single linkage is intended to catch elongated, rich-with-outliers clusters; Complete Linkage aims to catch more compact clusters (if any), while Ward's

method should create those of relatively uniform sizes to get a picture of more distinctive clusters.

By using all three linkage criteria, these paper produces a more comprehensive understanding of the data. If all three methods produced similar clusters, this could provide greater confidence in the results. Alternatively, if the methods produced different clusters, this could indicate that the data may have multiple, equally valid cluster solutions, or that certain features of the data are more pronounced under different criteria. In any case, using multiple linkage criteria allowed for a more robust and nuanced analysis of the data.

The data source is a set of the annual statistical reports by the National Statistical Bureau of the Agency for Strategic Planning and Reforms of the Republic of Kazakhstan. This is the main statistical agency of the republic, and often the only source of statistical data about the country, which is also referred to by international organizations like the World Bank. Before clustering, the data was standardized using z-scores to allow for comparison of indicators with different units of measurement. The cluster objects are 38 cities of republican and regional significance in Kazakhstan, of which 35 are "cities of regional significance" and 3 are "cities of republican significance".

There are five groups of factors to consider when designing a smart city. These include social, technological, economic, environmental, and political factors [1, 35]. The data used in the analysis includes indicators such as population growth, migration, education, employment, crime rate, pollution rate, etc. A complete list of the indicators used for the cluster analysis in this study can be found in the supporting information S1 Data. The selected set of indicators also determines the level of development of the city.

The significance of these variables has been emphasized by numerous authors. For example, Giourka et al. [36] have shown that population dynamics, such as population growth and urban migration, serve as significant drivers of urban expansion, necessitating the creation of efficient infrastructure and services to cater to the burgeoning populace. In the work, Selim [37] addressed challenges posed by aging infrastructure offering collaborative approaches like public-private partnerships, to navigate seamless integration of innovative solutions. The role of employment, education, and individual safety accentuating the allure of cities, promoting urban migration and fostering a conducive environment for residents were discussed by Napitupulu et al. [38]. Chrysostomou [39] highlighted the significance of healthy well-being, housing, entrepreneurship, tourism, and mobile working solutions contributing to the holistic development of urban spaces. Alnahari & Ariaratnam, [40] argued that technological and infrastructural advancements, including digital lifestyles, automation, smart infrastructure, e-mobility, and data connectivity, form the bedrock of the smart city framework. Suvarna et al. [41] have some evidence that the economic vitality of cities is bolstered by urban manufacturing and the pivotal role of small businesses. Chang et al., [42] stated that reshaping of urban landscapes is influenced by emphased inclusivity for women's economic contributions. Akimova et al. [43] investigated opportunities to use financial instruments such as subsidies to provide the impetus for technological adoption, while initiatives like retrofitting buildings emphasize sustainability and energy efficiency.

The analysis was performed with "R" programming language. The software used is "RStudio" expanded with "xlsx" [44], "clValid" [45]and "factoextra" [46] libraries. The "xlsx" library was used to import statistical data into RStudio environment as convenient data frames, "clValid" was used to validate the results of clustering and "factoextra" allowed for better visualisation of clustering results.

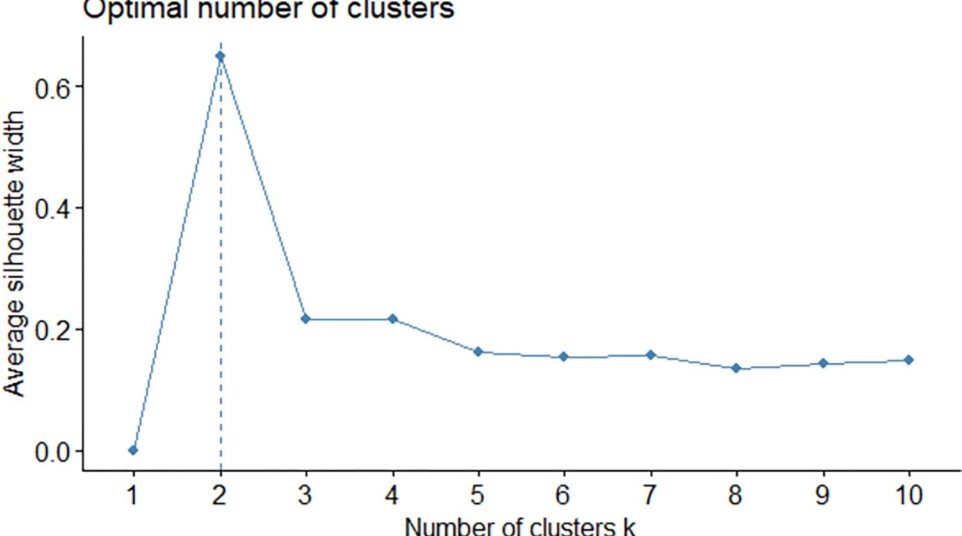

**Fig 1. Silhouette method results.**

## Results

The Silhouette method clearly shows that two clusters are the most justified number of clusters (Fig 1). However, to enrich the analysis, it seemed to be a good idea to use some other metrics to define the number of clusters.

To supplement the Silhouette method, another one was used, namely the so called "Elbow method". The results do not show a distinct fracture of the graph, thus giving some more freedom to use more clusters (Fig 2).

Given the results of the cutting criteria, a clustering was performed, using the abovementioned methods. To make two clusters, it was always sufficient to single out Almaty and Astana as the first cluster, and all other cities as the second one, with no difference caused by the

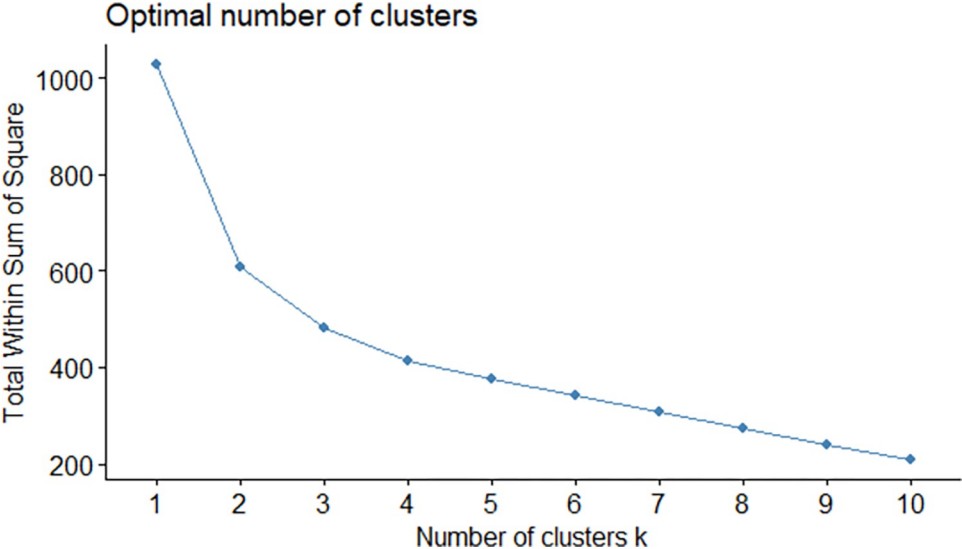

**Fig 2. Elbow method results.**

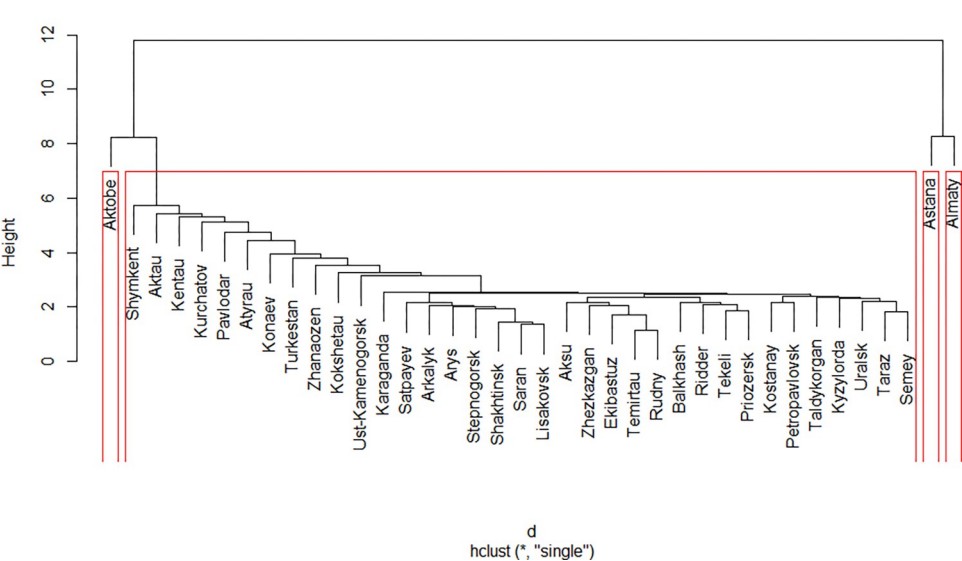

**Fig 3. Results of the Single Linkage clustering.**

change in clustering method. Therefore, they are passed directly to the interpretation section, without any dendrograms presented.

### Single Linkage

Single linkage clustering gives recognizable pattern, with the majority of cities being clustered consecutively in one big cluster, and others being distributed in smaller ones (Fig 3). In particular, cities of Almaty, Astana and Aktobe all have their own clusters. There was an option to make three clusters, after adding Aktobe to the biggest cluster, but that did not give any interesting results during the interpretation step, and thus was discarded.

### Complete Linkage

The Complete Linkage clustering created a different pattern, as expected, and allowed to distinguish between four clusters (Fig 4). Two of them emerged as expected, consisting of Almaty and Astana in the first one, and Aktobe on its own in the second one, whereas two remaining clusters have more or less even distribution of cities. In principle, it was possible to merge Aktobe and the fourth cluster, but with the results of the previous work on that matter [1], it seemed reasonable to leave the clusters as they are now, separate.

### Ward's method

The Ward's method gives the same number and content of the clusters as the Complete Linkage does, having the difference only in the sequence of merger of cities. Therefore, the interpretation of clusters for the Complete Linkage and Ward's method (Fig 5) was performed once, and the results are presented for both methods in the same table.

### Interpretation

There are three different interpretations, one for each of for the three different sets of clusters. The first table consists of two clusters, as the Silhouette method showed this to be the optimal

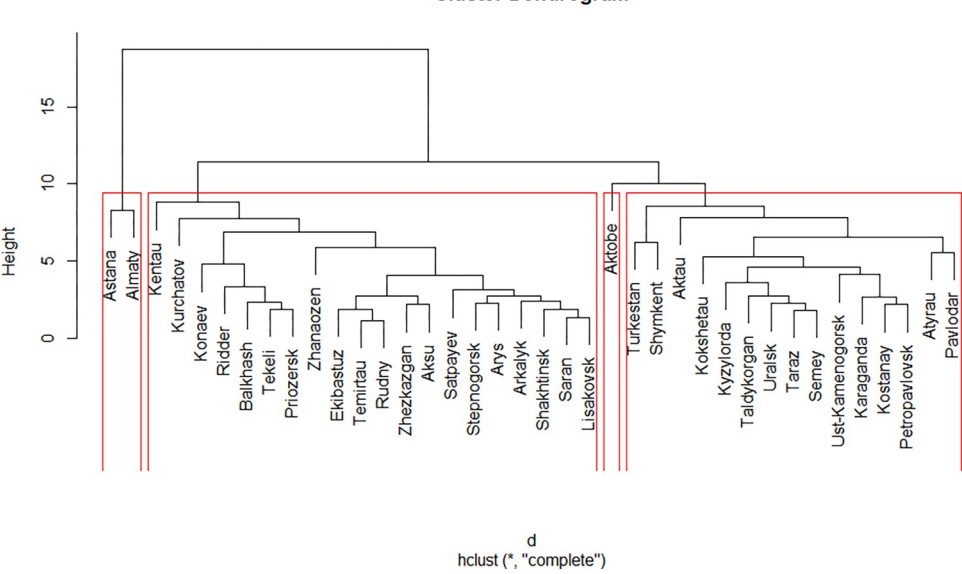

**Fig 4. Results of the Complete Linkage Clustering.**

number of clusters, and this could not be thoughtlessly ignored. The other two tables contain interpretations for the results of the Single Linkage and Complete Linkage / Ward's method respectively.

Having only two clusters might be beneficial, because it gives a cluster of (almost) absolute leaders, having the biggest averages of almost every positive variable (except "Hotels"). It is clearly seen that Almaty and Astana have the best performance regarding economic and technological development. They have their flaws, because of the biggest averages of emissions, unemployment and crimes, but the cumulative influence of the other variable should outweigh

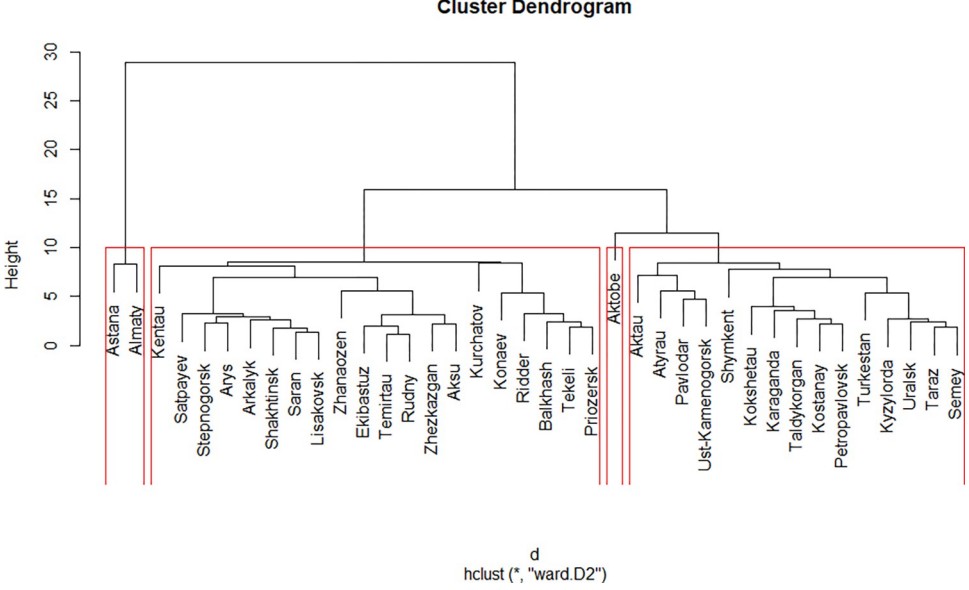

**Fig 5. Results of the clustering using Ward's method.**

**Table 1. Interpretation for 2 clusters.**

| Cluster | Max | Min | Interpretation |
|---|---|---|---|
| Cluster1: Almaty Astana | All others | Hotels | Leaders. These cities have the biggest potential to develop as smart cities and thus should be in the focus of policy. |
| Cluster2: All other cities | Hotels | All others | Followers. These cities should use the experience of the leaders to start developing smart cities. |

the disadvantages (Table 1). The cluster of all other cities, though being safer and more "hospitable", have hard times competing in other important aspects. This result adds little to the results of previous work other than a different set of variables for interpretation. However, due to the fact that one cluster is leading in all variables (except one), even this does not add noticeable insights.

Three of the four clusters created with the Single Linkage method are cities on their own, having their focal peculiarities. Astana is the city that is the quickest to grow in population. It is the most active in building new residential areas (and overhauling the old ones). It has the best production values, though paired with emissions, and its enterprises are more actively using the Internet. Given that, it has the greatest potential to use its growing population, economic prosperity to attract even more highly qualified people to grow as a smart city.

Almaty is the centre of higher education in Kazakhstan, and has the biggest population. Its business is the most active in using IT and automation, at the time being the most numerous, and welcoming to women as top-managers. This city gets the biggest amount of governmental investments, and the government is often welcome to have a share in the enterprises here. In other words, it is the city with the most developed infrastructure to become a smart city, has enough people and is able to educate them to support such a development. There are some problems, though. The city has the biggest crime rate, as well as unemployment.

Aktobe has some other advantages, namely, the highest number of enterprises with the foreign investments, with the lowest levels of crimes and unemployment. The problems are low tourism attractivity, low automation of enterprises, low living wage, and the least active residential overhauling among all clusters. In other words, the city is specialised on maintaining foreign business, that gives job places, but this business is quite conventional–e.g. natural resource extraction and preliminary processing.

The last cluster obtained contain all other cities, and has almost no advantages, except for the number of hotels, which cannot be useful without other factors to develop smart cities. Thus, the cities of the fourth clusters should be the followers of the three mentioned earlier, and use their results to decide which factors are more important to develop first to become smart cities.

The use of single linkage on a set of variables that was expanded compared to the previous article made it possible to divide Astana and Almaty into separate clusters and determine their features (Table 2).

The results of Complete Linkage and Ward's clustering have partly the same interpretation, as for Astana, Almaty (now being in one cluster), and Aktobe, but partly the different one, for two other clusters.

The first cluster containing Astana and Almaty has almost all the advantages of these two, thus being a cluster of the Cores of the Smart Cities. Their only "problematic" maximum is the maximum or crime rate, which is easy to expect from two biggest cities of Kazakhstan, since crime rates scale non-linearly with the city size [47].

**Table 2. Interpretation for Single Linkage clusters.**

| Cluster | Max | Min | Interpretation |
|---|---|---|---|
| Astana | Growth, Migration, Minimum, Residential, SMB, Ent_Internet, Ent_DSL, Production, Overhaul, Building, Emitters | None | Capital of Human Capital. Being attractive to internal migration means having access to human resources, which are the most valuable nowadays. |
| Almaty | Pop, Unemployment, Universities, Crimes, Computers, Ent_Automated, Invoices, Servers, Prod_Index, SB, Women, Ent_State, Investments | None | Development Nexus. Having well-developed infrastructure, innovative and numerous enterprises with dense population that has an easy access to education makes it the best first choice. |
| Aktobe | Ent_Foreign | Unemployment, Crimes, Minimum, Hotels, Ent_Automated, Overhaul, Ent_State | Safe Haven. Might use the access to foreign investments to improve other important aspects that might allow it to develop as a smart city. |
| Cluster4: All other cities | Hotels | Pop, Growth, Migration, Universities, Residential, SMB, Computers, Ent_Internet, Ent_DSL, Invoices, Servers, Production, Prod_Index, SB, Women, Ent_Foreign, Investments, Building, Emitters | Outsiders. These should be the last to develop as smart cities after the cities of the other clusters, and use their experience. |

The results for Aktobe do not differ much from those of the Single Linkage, except that here Aktobe has the biggest number of emitters, highlighting it as less healthy to live. In other words, more attention should be paid to promote ecologically safe environment for living in the city.

Cluster number three is interesting for two reasons. First, it contains mostly the cities that are the administrative centres of respective regions of the country, or are comparable to those. Second, they are average in every regard (except that they have the least production growth rate). These cities might follow the Cores and, probably, the Western Star, and form the second queue to develop as smart cities.

The fourth cluster consists of cities with the worst performance regarding chosen variables. They are mostly small cities, with the least number of emitters, thus with relatively healthy environment. These have the smallest potential to develop as smart cities, and will require the biggest amount of investments per capita. All clusters are presented in the Table 3.

The results show that using data for cities rather than regions of the country can identify heterogeneities in potential more accurately. In one region there may be cities from different clusters, for which it would be more reasonable to use different strategies for their development as smart cities. Expanding the list of variables for clustering made it possible to create a more detailed description of the clusters. In the future, this is an opportunity to implement targeted policies aimed at specific cities, rather than regions. The distribution of cities in the same region among different clusters, as well as the detailed characteristics of clusters, are the main contributions of this work.

## Discussion

Cluster analysis to assess the potential of smart cities in Kazakhstan has been used once before by Digel et al. [1]. There are similar studies for other countries, although some of them aim to cluster those cities that can be called smart to a certain degree. For example, Cantuarias-Villessuzanne et al. [16] used seven PCA ascending hierarchical classifications. They identified three clusters of cities in Europe: cities with new smart strategies, cities focused on technology, and smart cities focused on quality of life. The study focuses on cities with good development potential as smart cities and pays little attention to the rest. This makes sense on a European scale with many relatively developed cities. However, there are not that many well-developed (smart) cities in Kazakhstan to suffice for cluster analysis. Thus, the focus of present study was

**Table 3. Interpretation for Complete Linkage / Ward clusters.**

| Complete&Ward | Max | Min | Interpretation |
|---|---|---|---|
| Cluster 1:<br><br>Astana<br><br>Almaty | Pop, Growth, Migration, Universities, Crimes, Minimum, Residential, SMB, Computers, Ent_Internet, Ent_Automated, Ent_DSL, Invoices, Servers, Production, Prod_Index, SB, Women, Overhaul, Ent_State, Investments, Building | None | Cores of the Smart Cities. The most promising cities in terms of developing smart cities. |
| Cluster 2: Aktobe | Ent_Foreign, Emitters | Unemployment, Crimes, Minimum, Hotels, Ent_Automated, Overhaul, Ent_State | Western Star. Still should use the access to foreign investments to improve other important aspects that might allow it to develop as a smart city. Special attention should be paid to ecological issues. |
| Cluster 3: Kokshetau, Taldykorgan, Atyrau, Uralsk, Taraz, Karaganda, Kostanay, Kyzylorda, Aktau, Pavlodar, Petropavlovsk, Turkestan, Ust-Kamenogorsk, Semey, Shymkent | None | Prod_Index | Averages. Have no advantages nor disadvantages, thus being able to catch up and maintain the trend of developing smart cities, if it emerges. |
| Cluster 4:<br><br>Stepnogorsk, Konaev, Tekeli, Balkhash, Priozersk, Saran, Satpayev, Temirtau, Shakhtinsk, Arkalyk, Lisakovsk, Rudny, Zhanaozen, Aksu, Ekibastuz, Arys, Kentau, Kurchatov, Ridder | Hotels | Pop, Growth, Migration, Universities, Residential, SMB, Computers, Ent_Internet, Ent_DSL, Invoices, Servers, Production, SB, Women, Ent_Foreing, Investments, Building, Emitters | Outsiders. These have the lowest potential to develop smart cities, demanding more effort to start developing as smart cities. |

to cluster as many Kazakh cities as possible. As there are no smart cities in Kazakhstan, such an approach would be fruitless.

Using cities instead of regions to perform cluster analysis seems to be natural extension of previous research performed [1], sinceit is the cities, not the regions, that are in focus of smart development. The problem emerged in the realm of data availability, as some cities do not have data for some of the chosen variables, thus raising the dilemma of cutting the number of variables versus cutting the list of cities to analyse. This paper presents the second approach, but it could go the other way. Thus, here is the clustering of existing cities, but keeping this in mind, it is still possible to discuss the results and draw useful conclusions.

Going the easiest way and distributing available cities between two clusters gives the most obvious result. The two most developed cities in the country–Almaty and Astana–get their own cluster with almost all advantages, whereas other cities form the big cluster of "all others" remaining unsuitable to start developing as smart cities. They have great benefits, but turning them into smart cities can take a long time due to their scale. Moreover, understanding the characteristics of other cities is also necessary as they will also need to transform into smart cities later on.

Going the riskier way of getting clusters that are too much alike and distributing the cities between four clusters allows to obtain more detailed results. Apart from Almaty and Astana being special, such an approach highlights Aktobe as one of the cities with its own features, that might help or hinder its development as smart city, which is in line with the previous research. Using Single Linkage to cluster cities does not give much more information on top of already abovementioned, since all the cities except Almaty, Astana and Aktobe go to the cluster of "outsiders".

On the other side, using Complete Linkage or Ward's method give results worth of deeper investigation. Here, again, Almaty and Astana, as one, perform as the most suitable cores of the smart cities to start developing, having almost every advantage among present. It is

important for these cities not so much to create conditions for development as smart cities, but to weaken barriers. In particular, high crime rates indicate personal safety problems, while average emissions may indicate environmental problems. Creating a comfortable environment for living in smart cities is important, since they require a large number of highly qualified specialists to operate, and they are demanding of the quality of life in the city [48]. In addition, their size itself is a challenge that can only be overcome by competent management of the scaling of projects that can turn these cities into smart ones.

Aktobe does not have many strengths: a large number of enterprises with foreign participation, low crime and unemployment. The city is a leader in the level of emissions of harmful substances into the environment, has a low rate of housing construction and major renovations, low tourist attractiveness, and a level of automation. In other words, for its development it makes sense to take advantage of connections with foreign capital and innovation to solve environmental problems, as well as increase the pace of innovation and renovation of the housing stock.

The two other clusters, namely "Averages" and "Outsiders" also give some insights. "Averages" are the cities that do not perform particularly well, but neither bad. These could become the smart cities of the second queue, after Almaty, Astana and, possibly, Aktobe. The fact that many of these "Averages" are the centres of their regions supports this idea, since they could more easily spread their experience further using their administrative power. The main challenge for these cities will be the correct prioritization: what conditions to create and what difficulties to deal with first. Thus, the implementation of smart city projects in them primarily depends on the political will and quality of local government, which can competently plan their development.

The "Outsiders" cluster contain the cities that are still the least suitable to start smart city projects. They are the cities that would need the most investments per capita to develop smart cities. They are, however, usually smaller than the cities from other clusters, and thus might need lesser absolute amount of investments to start a smart city project. This might be used to test some project ideas, that could have costed more if started in a bigger city.

There are some problems and limitations present in this study. The most analytically justified number of clusters, namely two, is not very informative and interesting to derive some policy recommendations. However, making more cluster, although gives more information, creates less distinctive clusters, so that policy recommendations might not make much difference for cities from different clusters. Three of four clusters created by the Single Linkage method are just single cities known to be suitable for developing as smart cities. Thus, only the results of the Complete Linkage and Ward's method give distinctive and useful insights to include during policy development regarding smart cities.

There are some problems with data and variables. First, there are roughly half of the Kazakh cities present in the analysis. Though all big cities are included, there are still might be some small cities with distinctive feature missing due to lack of information for the analysis. Future research could try to address this feature. Another problem is difficulties to "translate" given factors and indicators into quantitative variables to be used for clustering. Some factors, like social and economic, are covered well, whereas technological and environmental are present only in half, and political being not covered almost at all. The biggest issue here is to find quantitative variables both present for cities of interest and being able to "proxy" the indicators, some of them described in ambiguous way. Search and addition of new quantitative variables to represent more indicators might significantly change the results of clustering, though it is assumed, that some most notable results will remain the same, namely the highest potential of Almaty and Astana, or the existence of "Averages". The main difference could emerge in the realm of recommendations, that could benefit a lot from more detailed set of indicators.

Future research could go the way of solving the abovementioned problems, as well as to make clustering inside certain regions to focus on their features, or on adapting existing findings to develop concrete policies.

## Conclusion

This study aimed to evaluate the features of the cities of Kazakhstan to highlight those having the best potential to develop as "smart cities." The main novelty of this work is the clustering of cities in Kazakhstan to assess their potential as smart cities, as well as the use of an expanded set of variables. This made it possible not only to confirm the conclusions of the previous study about the importance of Almaty, Astana and Aktobe as pioneers in this area, but also to identify a group of cities that could pick up this trend. In addition, preliminary recommendations for the development of smart cities in clusters were formed. The study supports the conclusion about deep regional inequality affecting the potential to successfully develop and manage smart cities. In addition to regional inequality, the analysis also revealed intra-regional differences. In particular, in the same region of the country there are cities belonging to different clusters, which can be used to develop targeted policies for the development of smart cities.

There are two cities with the highest potential–Almaty and Astana. They are the most developed with regards to human capital, infrastructure, education, IT, production, though being more dangerous. Aktobe city has a couple of distinctive features, like the highest share of enterprises with foreign shares, but is not as attractive as it seemed in the previous research. Some middle-sized cities might perform as followers to catch up the trend of developing smart cities after these two (three) start their smart city projects. The remaining cities will demand much more investments per capita to build smart cities and thus should not en masse be the first to implement the concept, but might serve as test sites for new ideas because of their small size. The best solution in the development of smart cities would be the launch of pilot projects in the small cities, with the most successful practices then scaled and implemented in the "Cores of the Smart Cities" and, maybe Aktobe, if it manages to use its advantages to compensate ecological risks.

## Supporting information

**S1 Data.**
(XLSX)

## Author Contributions

**Conceptualization:** Marat Urdabayev.

**Data curation:** Marat Urdabayev, Ivan Digel, Akan Nurbatsin.

**Formal analysis:** Anel Kireyeva, Ivan Digel.

**Funding acquisition:** Laszlo Vasa.

**Investigation:** Anel Kireyeva, Ivan Digel, Kuralay Nurgaliyeva.

**Methodology:** Marat Urdabayev, Kuralay Nurgaliyeva.

**Project administration:** Ivan Digel, Akan Nurbatsin.

**Resources:** Kuralay Nurgaliyeva.

**Software:** Akan Nurbatsin.

**Supervision:** Anel Kireyeva, Laszlo Vasa.

**Validation:** Anel Kireyeva, Laszlo Vasa.

**Visualization:** Akan Nurbatsin.

**Writing – original draft:** Marat Urdabayev.

**Writing – review & editing:** Laszlo Vasa.

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
