## [Decision Letter · Decision Letter 0]

21 Aug 2023

PONE-D-23-17894Discovering smart cities' potential in Kazakhstan: a cluster analysisPLOS ONE

Dear Dr. Vasa,

Thank you for submitting your manuscript to PLOS ONE. After careful consideration, we feel that it has merit but does not fully meet PLOS ONE’s publication criteria as it currently stands. Therefore, we invite you to submit a revised version of the manuscript that addresses the points raised during the review process.

ACADEMIC EDITOR:The novelty is poor in your paper, this point was strongly mentioned by reviewer #1. Thus, you should make sure to address the novelty clearly in the revised version of your paper. Provide novelty in your results as well. Additionally, address all the points that were made by the reviewers in step-by-step way, including analytical discussion.

We look forward to receiving your revised manuscript.

Kind regards,

Tarik A. Rashid, PhD

Academic Editor

PLOS ONE

“This research has funded by the Science Committee of the Ministry of Science and Higher Education of the Republic of Kazakhstan (Grant " Development Strategy of Kazakhstan Regional Potential: Assessment of Socio-Cultural and Economic Potentials, Roadmap, Models and Scenarios Planning " No. BR18574240).”

“This research has funded by the Science Committee of the Ministry of Science and Higher Education of the Republic of Kazakhstan (Grant " Development Strategy of Kazakhstan Regional Potential: Assessment of Socio-Cultural and Economic Potentials, Roadmap, Models and Scenarios Planning " No. BR18574240).”

“This research has funded by the Science Committee of the Ministry of Science and Higher Education of the Republic of Kazakhstan (Grant " Development Strategy of Kazakhstan Regional Potential: Assessment of Socio-Cultural and Economic Potentials, Roadmap, Models and Scenarios Planning " No. BR18574240).”

Reviewers' comments:

Reviewer's Responses to Questions

**Comments to the Author**

1. Is the manuscript technically sound, and do the data support the conclusions?

Reviewer #1: Partly

Reviewer #2: Yes

2. Has the statistical analysis been performed appropriately and rigorously? 

Reviewer #1: Yes

Reviewer #2: No

3. Have the authors made all data underlying the findings in their manuscript fully available?

Reviewer #1: Yes

Reviewer #2: Yes

4. Is the manuscript presented in an intelligible fashion and written in standard English?

Reviewer #1: Yes

Reviewer #2: Yes

5. Review Comments to the Author

Reviewer #1: The article describes the use of cluster analysis to identify cities in Kazakhstan with the highest potential for smart city development. It is a well-written article, providing a clear exposition of the context, methodology, and results, which can be easily understood by readers unfamiliar with the topic. However, there is no methodological innovation, and the results and their discussion do not contribute novelty to the current knowledge on the subject.

Regarding the methodology, an agglomerative cluster analysis is employed with three different forms of linkage between groups (Ward's criterion, Complete, and Single), applied to a dataset of 38 cities described by a set of 26 demographic, economic, and social variables. The methodology was previously used in a recent study, focusing on regions rather than cities, also in Kazakhstan. The material presented in Section 3 "Methods and Data" is standard material covered in a university course on data analysis or multivariate statistics.

The results of the three cluster analyses are disparate, but all of them indicate the existence of clusters formed by the two or three major cities, while the rest of the cities are agglomerated into a single cluster or two clusters. The conclusions drawn are overly generic, failing to fulfill the main objective of the article, which, in the authors' own words, is " to provide meaningful and actionable insights for policymakers and urban planners." The significance of the variables considered in the study and their influence in creating an appropriate environment for smart city development is not described. Therefore, the analysis and discussion of the results are superficial, and no conclusions are drawn from the data analysis that were not already known. Clearly, data analysis can be used to confirm existing knowledge; however, the conclusions should still be supported quantitatively by the results of the analyses.

Finally, the data contained in the supplementary Excel file includes anomalous values (e.g., a percentage of 45,000 in unemployment, and this is not the only case).

My recommendation would be for the authors to improve their variable dataset, discuss its relationship with smart city development, better justify the connection between the obtained clusters and the group's positioning towards the city's evolution into a smart city, and contrast the results with previous studies, either in the same geographical area or through similar analyses in other regions.

Reviewer #2: Title:

Discovering smart cities' potential in Kazakhstan: a cluster analysis

The manuscript is an interesting study on the potential for the development of smart cities in Kazakhstan.

However, there are some issues that can improve the manuscript:

It would be interesting to obtain more information on the type and quality of the data used for the study. A special mention should also be made and development in more depth to the origin of the data.

I have observed that the term “smart city” appears in the manuscript written in different ways, such as by "smart city" or smart city (with "" or without ""). The form of denomination should be unified. Perhaps the second option is more appropriate since the name smartcity is common in the field of research on the city.

The citations and bibliographical references must be reviewed. For example:

-The reference "Villessuzanne et al. (2021)" does not appear in the bibliography.

-The citation "(West, 2017)" appears in the text while the reference "West, G. (2018)" appears in the bibliography. It seems that one is wrong.

-In the text it mentions "Kubina and Vodak" (page 4) while in the bibliography it includes bibliographical reference "Kubina, M., Šulyová, D., & Vodák, J."

-In the text he cites "(Frazer, 2015)" twice, while in the bibliography only the reference "Fracer, 2019" appears.

- The reference "Héraud, J., & Muller, E. (2022)" appears in the bibligraphy. However, its corresponding citation does not appear in the text.

In the opinion of this reviewer, the manuscript meets the criteria of the journal and is suitable for publication with major revision.

6. PLOS authors have the option to publish the peer review history of their article (what does this mean?). If published, this will include your full peer review and any attached files.

Reviewer #1: No

Reviewer #2: No

---

## [Author Response · Author response to Decision Letter 0]

18 Nov 2023

Response to Reviewers

Reviewer #1: The article describes the use of cluster analysis to identify cities in Kazakhstan with the highest potential for smart city development. It is a well-written article, providing a clear exposition of the context, methodology, and results, which can be easily understood by readers unfamiliar with the topic. However, there is no methodological innovation, and the results and their discussion do not contribute novelty to the current knowledge on the subject. Regarding the methodology, an agglomerative cluster analysis is employed with three different forms of linkage between groups (Ward's criterion, Complete, and Single), applied to a dataset of 38 cities described by a set of 26 demographic, economic, and social variables. The methodology was previously used in a recent study, focusing on regions rather than cities, also in Kazakhstan. The material presented in Section 3 "Methods and Data" is standard material covered in a university course on data analysis or multivariate statistics. The results of the three cluster analyses are disparate, but all of them indicate the existence of clusters formed by the two or three major cities, while the rest of the cities are agglomerated into a single cluster or two clusters. The conclusions drawn are overly generic, failing to fulfill the main objective of the article, which, in the authors' own words, is " to provide meaningful and actionable insights for policymakers and urban planners." The significance of the variables considered in the study and their influence in creating an appropriate environment for smart city development is not described. Therefore, the analysis and discussion of the results are superficial, and no conclusions are drawn from the data analysis that were not already known. Clearly, data analysis can be used to confirm existing knowledge; however, the conclusions should still be supported quantitatively by the results of the analyses. Finally, the data contained in the supplementary Excel file includes anomalous values (e.g., a percentage of 45,000 in unemployment, and this is not the only case). My recommendation would be for the authors to improve their variable dataset, discuss its relationship with smart city development, better justify the connection between the obtained clusters and the group's positioning towards the city's evolution into a smart city, and contrast the results with previous studies, either in the same geographical area or through similar analyses in other regions. 

Dear Reviewer #1,

Thank you for your thorough review and constructive comments on our manuscript. We appreciate your acknowledgement of the article's clarity and exposition. Below, we address each of the concerns you raised.

1. Lack of Methodological Innovation and Novelty:

We understand the concern about the lack of methodological innovation. While the methodology used is not new, we argue that its application to the specific context of Kazakhstan is valuable, at least for the country’s policy-makers. However, the results obtained can also be used for comparison with city analyzes in other countries. This is not the purpose of this work, so it was not done.

2. Generic Conclusions and Lack of Actionable Insights:

We recognize the critique that the conclusions are overly generic. In the revised manuscript, we provided more detailed interpretation of clusters and possible ways of their development. We also elaborated on some variables that are of the most importance for certain clusters.

3. Influence of Variables on Smart City Development:

We acknowledge that the original manuscript did not thoroughly discuss the significance of each variable. In the revised version, we elaborated on this by detailing how each variable influences the potential for smart city evolution, and provided additional references supporting our choice of variables.

4. Comparisons with Previous Studies:

We have highlighted differences with previous research on the topic, but have not made further comparisons with work from other countries beyond what is already presented in the article.

5. Anomalies in the Supplementary Excel File:

We apologize for the oversight and thank you for pointing out the anomalous data values. This, however, did not affect the analysis in any way, since during the analysis the data set was checked for outliers and anomalies before calculations. They were corrected in the R dataframe, but not in the original table that was sent.

We hope that these revisions will address your concerns and improve the manuscript.

Reviewer #2: Title: Discovering smart cities' potential in Kazakhstan: a cluster analysis The manuscript is an interesting study on the potential for the development of smart cities in Kazakhstan. However, there are some issues that can improve the manuscript: It would be interesting to obtain more information on the type and quality of the data used for the study. A special mention should also be made and development in more depth to the origin of the data. I have observed that the term “smart city” appears in the manuscript written in different ways, such as by "smart city" or smart city (with "" or without ""). The form of denomination should be unified. Perhaps the second option is more appropriate since the name smartcity is common in the field of research on the city. The citations and bibliographical references must be reviewed. For example: -The reference "Villessuzanne et al. (2021)" does not appear in the bibliography. -The citation "(West, 2017)" appears in the text while the reference "West, G. (2018)" appears in the bibliography. It seems that one is wrong. -In the text it mentions "Kubina and Vodak" (page 4) while in the bibliography it includes bibliographical reference "Kubina, M., Šulyová, D., & Vodák, J." -In the text he cites "(Frazer, 2015)" twice, while in the bibliography only the reference "Fracer, 2019" appears. - The reference "Héraud, J., & Muller, E. (2022)" appears in the bibligraphy. However, its corresponding citation does not appear in the text. 

Dear Reviewer #2,

Thank you for your feedback on our manuscript titled "Discovering smart cities' potential in Kazakhstan: a cluster analysis." We appreciate your comments on the study's interest level and have taken your suggestions seriously. Below are our responses to each point you raised.

1. Type and Quality of Data:

We understand your argument that the manuscript lacked sufficient information regarding the type and quality of the data. We mentioned that the data was taken from the state statistics agency of Kazakhstan, and this is the only source of such extensive data, the quality of which is ensured by state standards. The data type is assumed to be quantitative, since the stated cluster analysis methods support only quantitative data. If we misunderstood your comment about "data type", please clarify what exactly is meant and we will make the necessary additions.

2. Consistency in the term “smart city”:

Thank you for pointing out the inconsistency in the terminology. We will standardize the term to "smart city" without quotation marks throughout the manuscript.

3. Citations and Bibliographical References:

We apologize for the inconsistencies and errors in the citations and references. We have corrected each point as follows:

• Added "Villessuzanne et al. (2021)" to the bibliography.

• Rectified the discrepancy between "(West, 2017)" in the text and "West, G. (2018)" in the bibliography.

• Unified the names for "Kubina and Vodak" in the text and "Kubina, M., Šulyová, D., & Vodák, J." in the bibliography.

• Corrected the citation "Frazer, 2015" to match with the available reference "Fracer, 2019" in the bibliography.

• Included a corresponding citation for "Héraud, J., & Muller, E. (2022)" in the text.

Thank you again for your careful reading and helpful suggestions. We believe that these changes will improve the quality and readability of the manuscript.

---

## [Decision Letter · Decision Letter 1]

18 Dec 2023

Discovering smart cities' potential in Kazakhstan: a cluster analysis

PONE-D-23-17894R1

Dear Dr. Vasa,

We’re pleased to inform you that your manuscript has been judged scientifically suitable for publication and will be formally accepted for publication once it meets all outstanding technical requirements.

Kind regards,

Dr. Rahul Priyadarshi

Academic Editor

PLOS ONE

---

## [Editor Report · Acceptance letter]

4 Mar 2024

PONE-D-23-17894R1 

PLOS ONE

Dear Dr. Vasa, 

I'm pleased to inform you that your manuscript has been deemed suitable for publication in PLOS ONE. Congratulations! Your manuscript is now being handed over to our production team.

Kind regards, 

on behalf of

Dr. Rahul Priyadarshi 

Academic Editor

PLOS ONE